# Effect of Curcumin Plus Piperine on Redox Imbalance, Fecal Calprotectin and Cytokine Levels in Inflammatory Bowel Disease Patients: A Randomized, Double-Blind, Placebo-Controlled Clinical Trial

**DOI:** 10.3390/ph17070849

**Published:** 2024-06-28

**Authors:** Amylly Sanuelly da Paz Martins, Orlando Roberto Pimentel de Araújo, Amanda da Silva Gomes, Fernanda Lívia Cavalcante Araujo, José Oliveira Junior, Joice Kelly Gomes de Vasconcelos, José Israel Rodrigues Junior, Islany Thaissa Cerqueira, Manoel Álvaro de Freitas Lins Neto, Nassib Bezerra Bueno, Marília Oliveira Fonseca Goulart, Fabiana Andréa Moura

**Affiliations:** 1Pós-Graduação da Rede Nordeste de Biotecnologia (RENORBIO), Universidade Federal de Alagoas (UFAL), Maceió 57072-970, AL, Brazil; amylly.martins@iqb.ufal.br; 2Instituto de Química e Biotecnologia (IQB/UFAL), Universidade Federal de Alagoas (UFAL), Maceió 57072-970, AL, Brazil; orlando.robertoceca21@professor.educ.al.gov.br (O.R.P.d.A.); islany.cerqueira@iqb.ufal.br (I.T.C.); 3Pós-Graduação em Nutrição (PPGNUT), Universidade Federal de Alagoas (UFAL), Maceió 57072-970, AL, Brazil; amanda.gomes@fanut.ufal.br (A.d.S.G.); jose.rodrigues@fanut.ufal.br (J.I.R.J.); nassib.bueno@fanut.ufal.br (N.B.B.); 4Faculdade de Nutrição (FANUT), Universidade Federal de Alagoas (UFAL), Maceió 57072-970, AL, Brazil; fernanda.araujo@fanut.ufal.br (F.L.C.A.); jose.junior@fanut.ufal.br (J.O.J.); joice.vasconcelos@fanut.ufal.br (J.K.G.d.V.); 5Pós-Graduação em Ciências Médicas (PPGCM), Universidade Federal de Alagoas (UFAL), Maceió 57072-970, AL, Brazil; mlinsneto@gmail.com; 6Programa de Pós-Graduação em Química e Biotecnologia (PPGQB), Universidade Federal de Alagoas (UFAL), Maceió 57072-970, AL, Brazil; 7Pós-Graduação em Ciências da Saúde (PPGCS), Universidade Federal de Alagoas (UFAL), Maceió 57072-970, AL, Brazil

**Keywords:** Crohn’s disease, ulcerative colitis, curcumin, black pepper, antioxidants

## Abstract

The development and course of inflammatory bowel disease (IBD) are significantly influenced by inflammation and oxidative stress. Antioxidant therapy is a promising therapeutic option to enhance the clinical results of these individuals in this particular scenario. The purpose of this study is to assess the impact of curcumin, with or without piperine, on cytokines, fecal calprotectin (CalF), and oxidative stress enzymatic and non-enzymatic indicators in patients with IBD. Methods: Patients with Crohn’s disease (CD) or ulcerative colitis (UC) who were at least 18 years old and had intact liver and kidney function participated in this randomized, double-blind trial (trial registration: ensaiosclinicos.gov.br as RBR-89q4ydz). For 12 weeks, participants were randomly assigned to one of three groups: placebo, curcumin (1000 mg/day), or curcumin plus piperine (1000 mg + 10 mg/day). In order to examine oxidative stress indicators, CalF, and pro-inflammatory cytokines, blood and fecal samples were obtained, both prior to and following the intervention time. Results: After adjusting for age, sex, and type of IBD, the curcumin plus piperine group had substantially higher serum levels of superoxide dismutase (SOD) than the placebo group (4346.9 ± 879.0 vs. 3614.5 ± 731.5; *p* = 0.041). There were no discernible variations between the groups in CalF, inflammatory markers, or other indicators of oxidative stress. Conclusions: In patients with inflammatory bowel disease (IBD), our study indicates that a 12-week curcumin plus piperine treatment effectively increases enzymatic antioxidant defense, especially SOD. These results demonstrate the potential therapeutic benefits of managing redox imbalance in individuals with IBD.

## 1. Introduction

Inflammatory bowel diseases (IBD), such as Crohn’s disease (CD) and ulcerative colitis (UC), are chronic and recurrent gastrointestinal disorders characterized by alternating periods of symptom activation and remission, including abdominal pain, diarrhea, and rectal bleeding. These conditions significantly compromise patients’ quality of life [1,2].

The number of new IBD cases is rising globally, according to epidemiological data, placing a significant strain on the healthcare system [3]. The cause of IBD is still not fully known, despite continued research. The onset and advancement of the inflammatory process are linked to the interplay of oxidative stress, an exaggerated, uncontrolled immune response, and hereditary and environmental variables [4].

In inflammatory bowel diseases (IBD), a redox imbalance occurs where the antioxidant defense is insufficient to counteract the excess of generated reactive oxygen and nitrogen species (RONS). This contributes to the impairment of the epithelial barrier and a consequent increase in intestinal permeability, allowing the entry of luminal antigens that induce a dysregulated immune response characterized by elevated reactive species and pro-inflammatory cytokines. This results in oxidative damage to cellular components, further exacerbating the chronic inflammatory state [5,6].

Currently, drug therapy of IBD involves synthetic drugs and monoclonal antibodies designed to manage imminent inflammation. However, prolonged use of these therapies may lead to adverse effects, limiting their effectiveness and causing many patients to become refractory to treatment, thereby increasing the risk of complications and surgical interventions [7,8].

To address these challenges, research has explored new therapeutic alternatives, with particular emphasis on the utilization of the plant *Curcuma longa*, commonly known as turmeric, and the three main curcuminoids—curcumin, demethoxycurcumin, and bisdemethoxycurcumin—of which curcumin is the main bioactive component [9].

Curcumin, a hydrophobic polyphenol, exhibits potent antioxidant, anti-inflammatory, antitumor, antimicrobial, anti-glycating, anti-coagulant, and healing properties, as evidenced by both experimental studies and clinical trials in IBD [10] and other clinical situations such as Type 2 diabetes [11] and some cancers [12]. However, due to reduced solubility, fast metabolization, and systemic elimination, leading to low bioavailability, its therapeutic potential is limited [13,14].

Piperine, a bioactive compound from black pepper (*Piper nigrum*), emerges as a potentiation agent capable of inhibiting hepatic glucuronidation, thereby increasing curcumin bioavailability by up to 2000% [15]. The combination of curcumin and piperine has shown successful outcomes in diseases where oxidative stress is a significant etiological factor, such as metabolic syndrome [16,17]. Despite the positive effects of curcumin or turmeric in IBD, the combined impact with piperine, along with its antioxidant effects and influence on inflammatory markers like fecal calprotectin and cytokines, remains unexplored in clinical trials [18]. This highlights a crucial gap in understanding the comprehensive effects of the mixture in patients with IBD.

Therefore, the objective of the present study was to clinically investigate the efficacy of using curcumin alone or in combination with piperine on markers of oxidative stress and inflammation in patients with IBD, in different matrixes (blood and feces).

## 2. Results

### 2.1. Baseline Characteristics

The present study was carried out between July 2021 and March 2023. Initially, 58 patients were enrolled, with 51 completing the 12-week supplementation period: 14 in the placebo group, 20 in the curcumin group, and 17 in the curcumin + piperine group. Participants were excluded due to loss of contact during follow-up, voluntary discontinuation of the supplement, and incorrect intake of the recommended dosage (Figure 1).

The participants’ mean age was 47.5 ± 15.5 years, with the majority being female (*n* = 38; 65.5%) and in a stable relationship (*n* = 44; 75.9%). None of the patients consumed alcoholic beverages, and physical activity was not a prevalent practice. Changes in nutritional status among IBD patients are correlated with a high prevalence of overweight (*n* = 25; 43.1%) and a lack of activity.

Since biologic therapy is more effective in controlling the inflammatory condition, it became the main pharmacological treatment among study participants (*n* = 30; 51.7%). Notably, the results observed in Table 1 highlight a low prevalence of extraintestinal symptoms (*n* = 5; 8.6%) and non-communicable chronic disorders (*n* = 17; 29.7%).

The curcumin group had more patients with ulcerative colitis (*p* = 0.026) and the placebo group had a significantly greater prevalence of education beyond 4 years (*p* = 0.031), notwithstanding the randomization that was done at the start of the clinical study. By the end of the experiment, though, these differences were no longer statistically significant.

#### 2.1.1. Effects of Supplementation on Gastrointestinal Symptoms

In this study, as a secondary outcome, we assessed the impact of supplementation with curcumin, with or without piperine, on various gastrointestinal complaints common in patients with IBD. Initially, 41.4% (*n* = 24) of the participants reported heartburn. By the end of the study, this number had decreased to 23.5% (*n* = 12), with a notable decline in the placebo group, where heartburn complaints dropped from 31.6% (*n* = 6) at baseline to 7.1% (*n* = 1) at the study’s completion. Although there was a reduction in the number of participants in the curcumin + piperine group experiencing heartburn by the end of the study (47.1%, *n* = 8) compared to the baseline (57.9%, *n* = 11), this group still exhibited significantly higher complaints of heartburn compared to the placebo and curcumin groups. Specifically, at the study’s conclusion, 47.1% (*n* = 8) of participants in the curcumin + piperine group reported heartburn, compared to 7.1% (*n* = 1) in the placebo group and 15.0% (*n* = 3) in the curcumin group. This observed increase in heartburn in the curcumin + piperine group was statistically significant, as demonstrated by the chi-square test (*p* = 0.017) and GEE analysis (*p* = 0.016). The odds ratio (OR: 4.151; 95% CI: 1.309–13.169) indicates that participants receiving curcumin + piperine were 4.1 times more likely to report heartburn compared to the placebo group. However, this statistical significance is largely attributable to the marked improvement in heartburn symptoms in the placebo group, rather than an increase in complaints among those taking the supplement. Thus, while the combination of curcumin + piperine appears effective in enhancing antioxidant effects, the observed increase in heartburn reports should be interpreted in the context of overall symptom improvement in the placebo group, highlighting a need for cautious evaluation in clinical practice (Appendix A).

#### 2.1.2. Effects of Supplementation on Redox Imbalance and Inflammation

Among the oxidative stress and inflammation markers (Table 2 and Figure 2) evaluated in this study, supplementation with curcumin + piperine for 3 months proved effective in significantly increasing SOD levels compared to the placebo group after adjusting for sex, age, and type of IBD (4346.9 ± 879.0 vs. 3614.5 ± 731.5; *p* = 0.02; CI95%: 102.262–1528.186). Furthermore, it demonstrated a more pronounced impact when comparing the endline to the baseline (Δ) (538.8 ± 1040.1 vs. −126.8 ± 762.7, placebo group; *p* = 0.027; CI95%: −351.986–1094.626). The other redox markers (CAT, MDA, and H_2_O_2_ levels), disease activity (CalF) (Figure 1), inflammation (MPO, TNF-α, IL-17A, and IL-22), and anti-inflammatory markers (IL-10) did not show significant changes after the experimental period. All together, these findings highlight curcumin’s antioxidant effectiveness, which is especially noticeable in the rise in SOD levels, the body’s main defense against reactive oxygen species (ROS). However, the three-month treatment time was insufficient to modify the other biomarkers that were investigated.

Originally estimated for 20 people per group, the final analysis of SOD levels was based on 51 participants. This sample size reduction diminished the study’s statistical power. With a pooled standard deviation of 809 U/L, the average SOD levels at the conclusion of the intervention were 3614.5 U/L, 3910.2 U/L, and 4346.9 U/L for the placebo, curcumin, and curcumin + piperine groups, respectively. As a result, the study’s statistical power of 59.1% suggests that participant dropout may have contributed to the study’s relative underpowering, which may have impacted its capacity to identify meaningful differences between the groups.

## 3. Materials and Methods

### 3.1. Study Type and Design

This randomized, double-blind, placebo-controlled clinical study was conducted at the Coloproctology sector of Hospital Universitário Professor Alberto Antunes, Maceió-AL, Brazil, from July 2021 to March 2023.

The participants who met the eligibility requirements were those who had been diagnosed with mild to moderate CD or UC, were at least 18 years old, were receiving conventional medication therapy for IBD (aminosalicylates, immunosuppressants, or biological therapy), or who were not on a specific medication prescription for IBD (aminosalicylates, immunosuppressants, corticosteroids, or biological therapy), and had renal and liver function preserved.

Exclusion criteria encompassed individuals with one or more of the following conditions: (1) fistulas, abscesses, and strictures; (2) extensive resections of the small intestine (<200 cm of remaining intestine); (3) patients admitted for acute treatment, in serious general condition; (4) cancer; (5) HIV positivity; (6) pregnant and breastfeeding women; and (7) hospitalization for IBD in the last three months. Exclusionary factors of the clinical trial were: (1) gestation; (2) change or withdrawal of conventional dosage/medication for IBD; (3) HIV-positive patients and compromised renal and liver functions; (4) patients who voluntarily discontinued supplementation; and (5) patients who were hospitalized for IBD during the clinical trial. All participants signed an Informed Consent Form.

This study was registered at ensaiosclinicos.gov.br as RBR-89q4ydz on 20 July 2023. It was approved by the Ethics Committee of the Federal University of Alagoas under the process no. 7829516.5.0000.5013 on 28 February 2019.

### 3.2. Intervention

Participants who adhered to the clinical therapeutic regimen provided by a healthcare provider in the sector maintained their clinical therapeutic regimen, and were given prescriptions for curcumin powder (at doses of 1000 mg/day) and piperine (10 mg/day). The prescribed amounts were encapsulated into two carefully crafted pills, to reduce color distortion.

Patients were instructed to take these capsules after lunch consistently over a 12-week period. The co-administration of curcumin (doses of 1000 mg/day) or curcumin + piperine with lipid-containing meals was strategically employed to improve the bioavailability of these supplements, thereby facilitating optimal absorption.

The fact that these capsules were gluten- and lactose-free and gastro-resistant is notable. The selected doses of curcumin and piperine was determined by the amount that is deemed safe for humans (up to 6000 mg) and by the results of a double-blind, randomized trial performed by Petterson et al. (2018), which assessed the impact of this combination on the microbiota of healthy people [19].

Depending on the treatment group, an external compounding pharmacy produced the capsules using either piperine + curcumin or curcumin powders (made with gastroresistant capsules, free of lactose and sugar). To avoid bias, the placebo capsules, which were made of starch and orange food coloring, exactly matched the treatment capsules in terms of color, texture, and look. The supplementation regimen followed the guidelines set forth by the hospital’s monitoring and medication delivery department, guaranteeing the patients’ consistent experience.

### 3.3. Ingredient Characterization

The curcumin and piperine powders were obtained from the company Fragon^®^ with curcumin making up a total of 98% of the curcuminoids (leaflet included). HPLC analysis was performed as previously described [20], with slight modifications. Curcumin and piperine powders (0.25 mg/mL) and a combination thereof (9.9:0.1 *w*/*w*), diluted in acetonitrile, were used to obtain the chromatograms. The mobile phase was composed from deionized water, acidified with phosphoric acid (0.1% *v*/*v*) (solvent A) and acetonitrile (solvent B). The chromatograms were obtained with phase A + B (30:70 *v*/*v*), in 9 min, with a flow of 1.0 mL/min and temperature of 35 °C, and chromatograms were recorded at 345 nm (Appendix A).

### 3.4. Blood Collection and Sample Preparation

Participants’ blood was collected in tubes containing EDTA and centrifuged at 4000× *g* at 4 °C for 10 min. Subsequently, the plasma was aspirated and stored in aliquots at −80 °C in a biofreezer for the analysis of inflammatory and oxidative stress markers (primary outcome).

### 3.5. General Data and Anthropometric Measurements

The following information was gathered using a personalized questionnaire:(a)Socioeconomic information:
education: less than four years (corresponding to an incomplete primary education) or more than four years;self-declared racial categories: white, black, brown, indigenous, and yellow;marital status: either in a stable relationship or not (divorced, widowed, or single).(b)Clinical background:
nature of IBD and the date of diagnosis;existence of extraintestinal manifestations; and usage of certain medications for IBD treatment or not;historical information regarding COVID-19 infections was gathered.(c)Lifestyle:alcoholism and smoking;consistent exercise: physical activity was deemed consistent when it was done to preserve or enhance physical capacity.(d)Symptoms related to the stomach (secondary outcome)
Anthropometric assessment: Performed by a qualified expert prior to and following the intervention period, it involved weight (kg) and height (m) measurements to determine the body mass index (BMI), which was expressed in kg/m^2^. The appropriate cutoffs were used according to age: adults [21] and elderly people [22].

### 3.6. Fecal Calprotectin Analysis (CalF)

CalF levels were assessed using Bühlmann’s Quantum Blue^®^ (Schönebuech, Switzerland) test, according to the manufacturer’s information. CalF values were considered elevated when reaching ≥ 200 μg/g of feces, indicative of active disease.

### 3.7. Inflammation and Oxidative Stress Analyses

Malondialdehyde (MDA) levels were measured using reversed-phase ion-pair high performance liquid chromatography (HPLC) with ultraviolet detection at 270 nm, expressed as ng/µL [23]. Superoxide dismutase (SOD) activity was quantified using the SOD Assay kit from Sigma—WST^®^ (Sigma Aldrich, St. Louis, MO, USA), per the manufacturer’s instructions, measured at 450 nm absorbance in a spectrophotometer, with the results expressed in U/µL. Hydrogen peroxide (H_2_O_2_) was measured according to the protocol established by Pick and Keisari (1980), and the results read at 610 nm and expressed as ng/µL [24]. Catalase (CAT) activity was determined following the colorimetric method adapted by Aebi (1974), monitored at 240 nm and expressed in U/µL [25].

Myeloperoxidase (MPO) activity was measured by adapting the method previously proposed by Bradley et al. (1982), and the results were expressed in U/µL. One unit of MPO was defined as the amount of enzyme required to decompose 1 μmol H_2_O_2_ [26].

Levels of tumor necrosis factor alfa (TNF-α), interleukin 17A (IL-17A), IL-22, and IL-10 were determined by enzyme-linked immunosorbent assay (ELISA), using the PeproTech^®^ kit (PeproTech Brasil FUNPEC, Ribeirão Preto, SP, Brazil),according to the manufacturer’s instructions. Absorbance was read at 450 nm in an ELISA plate reader, and the results were expressed in pg/µL.

Furthermore, as a secondary outcome, the effect of curcumin supplementation, with or without piperine, was assessed concerning the primary gastrointestinal symptoms in patients with IBD.

### 3.8. Sample Size

The choice to investigate the effects of supplementing with curcumin on SOD levels was informed by research that our group has done, particularly with animal models. The rationale behind this decision was the paucity of human studies—both non-RCT and RCTs—that evaluated the impact of curcumin or curcumin + piperine on oxidative stress indicators in patients with inflammatory bowel disease (IBD). The relevant experimental clinical trial, which served as the foundation for the sample size calculation, can be found at https://www.repositorio.ufal.br/handle/riufal/5281 (accessed on 11 June 2024), the Universidade Federal de Alagoas’ thesis repository. A minimum sample size of 19 patients per group was determined by taking these data into consideration and taking into account the patient population at the Hospital Universitário Professor Alberto Antunes (HUPAA) Coloproctology outpatient clinic, to achieve 80% power for detecting a significant difference between groups at a 5% significance level.

### 3.9. Randomization, Blinding and Allocation

Participants were randomized using the ‘runif’ function of the “R” software (v. 4.1.3, The R Project Team, Vienna, Austria). A sequence of random numbers between 0 and 2 was requested to generate the list with a random sequence. This list was maintained by a researcher who had no contact with the participants. They were included sequentially to guarantee the allocation concealment. Rounding the generated numbers determined group assignments: participants assigned “0” were placed in the placebo group, those assigned with the number “1” in the curcumin experimental group, and those assigned with the number “2” in the curcumin + piperine experimental group. An independent evaluator, who remained blinded to both the participants and researchers throughout the study, was responsible for conducting the statistical analyses.

### 3.10. Statistical Analysis

All analyses were conducted using the Statistical Package for Social Science (SPSS^®^), version 26.0 (SPSS, Chicago, IL, USA). The descriptive results were expressed as mean and standard deviation (SD) or median (interquartile range), and frequencies (n and %). Variance homogeneity was assessed using Levene’s test. Variables that were not homogeneous were logarithmized. Delta (Δ = endline [T2] minus baseline [T1]) was calculated only on homogeneous variables. The comparison of categorical outcomes between treatment groups was made using the chi-square test/Fisher’s exact and generalized estimating equations (GEE), adjusted by sex, age, and IBD type. For continuous outcomes, an Analysis of Covariance (ANCOVA) was performed, followed by the Bonferroni test, adjusting for sex, age, and type of IBD. The results were considered statistically significant with a *p*-value < 0.05.

## 4. Discussion

This is, as far as we are aware, the first randomized, double-blind clinical research evaluating the effects of curcumin supplementation on oxidative, inflammatory, and disease activity markers in patients with inflammatory bowel disease (IBD), either on its own or in conjunction with piperine. Based on a notable increase in serum SOD levels among supplemented patients over the course of the three-month intervention period, our findings allow us to confirm that the standard commercially available powder, rich in curcumin and combined with piperine, exhibits significant antioxidant action by enhancing the endogenous defense mechanism. These novel findings for this polyphenol highlight its advantageous benefits for IBD patients. Our study group’s latest meta-analysis [10] provides more support for this.

The SOD family in mammals comprises three isoforms: SOD1 (Cu/ZnSOD, cytosolic), SOD2 (MnSOD, mitochondrial), and SOD3 (Cu/ZnSOD, extracellular). Their primary role is to scavenge superoxide radicals (O_2_^−•^), reactive signaling molecules that can induce cell damage. SODs convert O_2_^−•^ into H_2_O_2_, a less reactive form that can be further detoxified by catalase or glutathione peroxidase (GPx) [27,28]. Additionally, SOD activity reduces iron release, inhibiting hydroxyl radical (^•^OH) formation by means of Fenton reaction, and its downstream effects on lipids, proteins, and DNA [29].

On the other hand, lowering the amount of hydroxyl radical in the blood influences the production of reactive nitrogen species (RNS). This is because the superoxide radical anion can immediately combine with nitric oxide (^•^NO) to generate peroxynitrite (ONOO^−^), which is a strong oxidant and nitrating agent that can react with a variety of macromolecules. Furthermore, ONOO^−^ can diffuse across multiple cell diameters and is relatively persistent in physiological settings, which increases its toxicity [28]. In this case, the rise in SOD levels not only reduces the synthesis of peroxynitrite, which also has an impact on macromolecules, but also makes it possible to use nitric oxide as a vasodilator, which is crucial in a variety of biological circumstances.

These results are corroborated by a randomized clinical trial in patients with metabolic syndrome, where the combination of curcumin plus piperine, supplemented at the same dosage used daily in this study (1000 mg + 10 mg) for 8 weeks, significantly increased SOD activity [30]. Another study conducted by Panahi et al. [31] also demonstrated the effectiveness of the Curcumin C3 Complex^®^ formulation (1500 mg/day) in increasing SOD activity in patients with osteoarthritis after 6 weeks [32], thus demonstrating the effectiveness of curcumin on this marker of redox imbalance.

The observed increase in SOD levels in this study among patients receiving curcumin plus piperine, with no alterations in H_2_O_2_ and MDA levels, suggests a beneficial effect of this supplementation. It is noteworthy that oxidative stress in IBD is considered both an etiological factor and a trigger for exacerbations, negatively impacting the quality of life of individuals with IBD. Studies indicate significant changes in SOD activity among IBD patients compared to individuals without the disease [33]; these low levels are associated with an increase in the inflammatory process [34]. Another change found in those individuals pertains to the expression of all three isoforms of SOD in the colonic mucosa. Kruidenier et al. [35] concluded in their study with 29 IBD patients that both CD and UC are accompanied by increased intestinal Mn-SOD and decreased Cu/Zn-SOD and EC-SOD levels, particularly in the inflamed mucosal epithelium.

In this context, it becomes clear that finding strategies, whether pharmacological or not, to improve the activity of this enzyme is crucial to the clinical therapy of patients. The use of SOD or SOD mimetics has been tested as a pharmacological strategy in various in vivo and in vitro models [36], including induced colitis either through direct administration [37] or by stimulating the production of this enzyme in *Lactobacillus* [38]. However, these interventions are not yet considered a viable/safe option in humans; hence, randomized clinical trials testing these formulations are not readily available. Considering this, antioxidant therapy with natural or synthetic substances has been the subject of numerous investigations.

Recently, in a meta-analysis published by our group evaluating clinical trials that investigated antioxidant or anti-inflammatory actions in their treatments for IBD, only SOD was significantly modulated, while markers such as MDA, like our findings, and total antioxidant capacity showed no significant relevance [18].

As mentioned earlier, despite curcumin being a widely studied polyphenol in IBD, its antioxidant and anti-inflammatory roles have only been confirmed in animal models of ulcerative colitis and not in clinical trials, underscoring the novelty of our results.

Regarding the impact of supplementation on the inflammatory profile, in the present study, no significant changes were found in the plasma levels of the investigated markers (TNF-α, IL-22, IL-17, IL-10, and MPO), nor in CalF, reflecting the inflammatory activity of the intestine. These results align with a study by Sadeghi et al. (2020), conducted in patients with UC, where the administration of 1500 mg/day of curcumin for 8 weeks did not cause changes in TNF-α levels [39].

Notwithstanding the encouraging results from our study, it is important to address some limitations, such as the loss of five patients in the placebo group, the short duration of supplementation (3 months), and the inclusion of patients from a single follow-up center; it is crucial to emphasize that both treatment groups remain representative in relation to the calculated sample size. The brief supplementation period was strategically based on the standard interval between medical consultations in the department itself, and the single follow-up center serves as the state reference for patients with IBD.

Additionally, we recognize the limitation regarding the absence of mucosal inflammation markers. Studying these markers would require performing biopsies, which are not routine procedures for patients with IBD. It would not be ethical to request such invasive tests unless medically necessary.

Compared to the placebo group, the curcumin plus piperine group had a higher incidence of heartburn; this did not result from an increase in cases within the supplemented group, but rather a decrease in heartburn cases in the placebo group. Moreover, no participants needed to discontinue the supplementation due to heartburn, indicating good overall tolerance. This side effect may be attributed to the inhibitory effects of curcumin on cyclooxygenase-1 and -2, reducing prostaglandin synthesis which protects the gastric mucosa. Despite these effects, the safety of curcumin and piperine supplementation remains evident as no serious adverse effects were reported [40,41].

Together, these results underscore the necessity of a multifaceted treatment approach for IBD, given its complex and multifactorial nature. Antioxidant therapy aims to complement conventional treatments by improving the quality of life for patients. The combination of curcumin and piperine appears to effectively target specific aspects of IBD, particularly by controlling superoxide radicals and minimizing the production of reactive nitrogen species. This highlights its potential as a supportive therapy without significant adverse effects. Further studies on curcumin, including those involving patients with active disease, are warranted to fully understand its metabolic effects. Such research will ensure that curcumin can be safely and effectively prescribed by healthcare professionals treating individuals with IBD.

## 5. Conclusions

In summary, this result offers new insights into the use of curcumin, especially in combination with piperine, for enhancing antioxidant effects in individuals with IBD. The results provide new insights into the use of a natural therapeutic strategy and underscore the effectiveness of its combination with piperine in enhancing the antioxidant effects of curcumin. This approach presents therapeutic alternatives with minimal adverse effects, expanding the possibilities for individualized treatment and supporting the effective continuity of the treatment.

## Figures and Tables

**Figure 1 pharmaceuticals-17-00849-f001:**
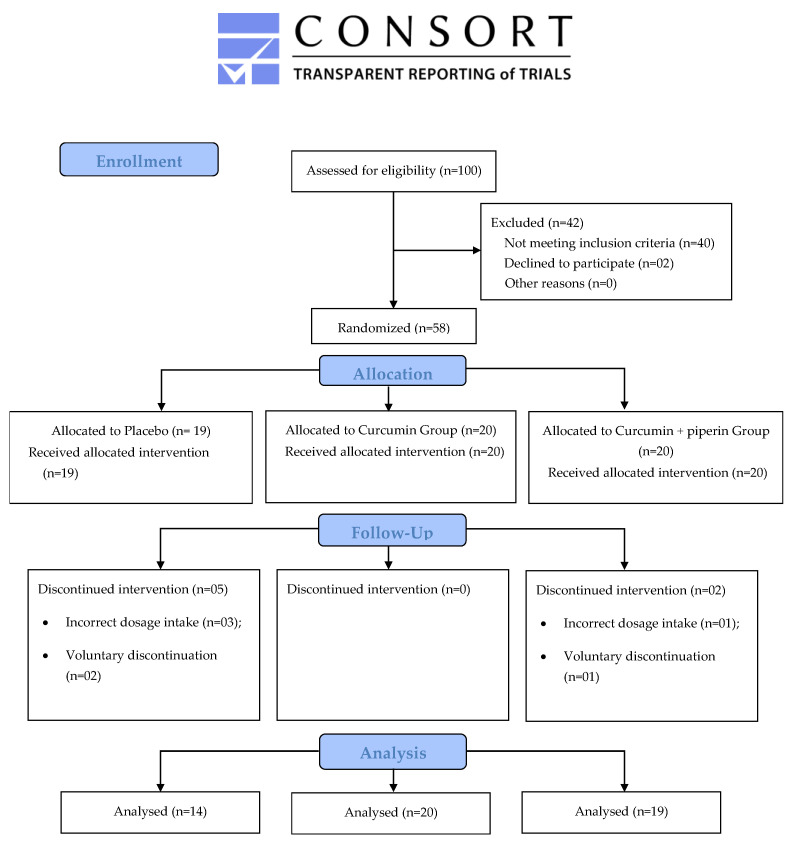
Flowchart of the clinical study.

**Figure 2 pharmaceuticals-17-00849-f002:**
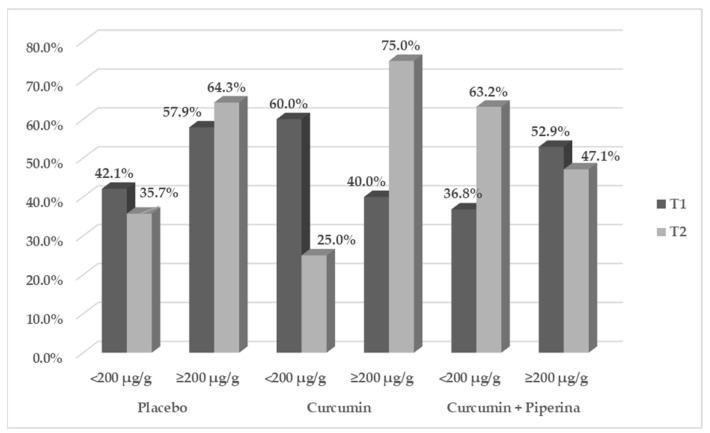
Fecal calprotectin of patients with inflammatory bowel disease according to the treatment group: baseline data (T1) and endline data (T2). Legend: Data expressed as %; GEE: generalized estimating equations-group vs. time interaction, adjusted by sex, age, and inflammatory bowel disease type: placebo group vs. curcumin + piperine group: *p* = 0.544 (OR = 1.392; 95% CI = 0.479–4.049); placebo group vs. curcumin group: *p* = 0.710 (OR = 1.231; 95% CI = 0.413–3.670).

**Table 1 pharmaceuticals-17-00849-t001:** Baseline Demographics and Clinical Characteristics of the IBD Patients.

	Total		Groups			*p*-Value
			Placebo *n* = 19	Curcumin *n* = 20	Curcumin + Piperine *n* = 19	
Age	Mean (SD)	47.5 ± 15.5	50.9 ± 14.4	46.9 ± 18.9	44.7 ± 12.4	0.460 ^1^
Sex	Female	38 (65.5)	13 (68.4)	11 (55.0)	14 (73.7)	0.447 ^2^
	Male	20 (34.5)	6 (31.6)	9 (45.0)	5 (26.3)	
Schooling	<4 years	38 (65.5)	8 (42.1)	15 (75.0)	15 (78.9)	0.031 ^2^
	≥4 years	20 (34.5)	11 (57.9) ^a^	5 (25.0) ^b^	4 (21.1) ^b^	
Self-declared race	White	14 (24.1)	2 (10.5)	7 (35.0)	5 (26.3)	0.196 ^2^
	Black/Brown	44 (75.9)	17 (89.5)	13 (65.0)	14 (73.7)	
Marital status	Single/Divorced	19 (32.8)	6 (31.6)	8 (40.0)	5 (26.3)	0.668 ^2^
	Stable union	33 (67.2)	10 (52.6)	10 (50.0)	13 (68.4)	
Inflammatory Bowel Disease	Crohn’s disease	19 (32.8)	9 (47.4) ^b^	2 (10.0) ^a^	8 (42.1) ^ab^	0.026 ^2^
	Ulcerative colitis	39 (67.2)	10 (52.6)	18 (90.0)	11 (57.9)	
Diagnosis time	<10 years	22 (37.9)	11 (57.9)	5 (25.0)	6 (31.6)	0.084 ^2^
	≥10 years	36 (62.1)	8 (42.1)	15 (75.0)	13 (68.4)	
Chronic non-communicable diseases	No	41 (70.7)	14 (73.7)	14 (70.0)	13 (68.4)	0.935 ^2^
	Yes	17 (29.3)	5 (26.3)	6 (30.0)	6 (31.6)	
History of COVID-19	No	47 (81.0)	17 (89.5)	15 (75.0)	15 (78.9)	0.495 ^2^
	Yes	11 (19.0)	2 (10.5)	5 (25.0)	4 (21.1)	
Extraintestinal manifestation	No	53 (91.4)	17 (89.5)	19 (95.0)	17 (89.5)	0.776 ^2^
	Yes	5 (8.6)	2 (10.5)	1 (5.0)	2 (10.5)	
Pharmacologic treatment	Aminosalicylates	24 (41.4)	6 (31.6)	8 (40.0)	10 (52.6)	0.520 ^2^
	Immunosuppressant alone/combined with aminosalicylates	3 (5.2)	1 (5.3)	2 (10.0)	0 (0.0)	
	Biological therapy	30 (51.7)	11 (57.9)	10 (50.0)	9 (47.4)	
	No drug therapy	1 (1.7)	1 (10.5)	0 (0.0)	0 (0.0)	
Smoking	No	43 (74.1)	13 (68.4)	15 (75.0)	15 (78.9)	0.755 ^2^
	Yes	15 (25.9)	6 (31.6)	5 (25.0)	4 (21.1)	
Physical exercise	No	42 (72.4)	13 (68.4)	15 (75.0)	14 (73.7)	0.890 ^2^
	Yes	16 (27.6)	6 (31.6)	5 (25.0)	5 (26.3)	
Body mass index	Low weight	9 (15.5)	1 (5.3)	4 (20.0)	4 (21.1)	0.588 ^2^
	Suitable weight	24 (41.4)	10 (52.6)	7 (35.0)	7 (36.8)	
	Overweight	25 (43.1)	8 (42.1)	9 (45.0)	8 (42.1)	

Legend: ^1^—ANOVA test; ^2^—Chi-square or Fisher’s exact test; Categorical data expressed as n (%); ^a,b^: different letters represent statistical significance.

**Table 2 pharmaceuticals-17-00849-t002:** Serum Levels of Oxidative and Inflammatory Biomarkers in Patients with Inflammatory Bowel Disease According to Treatment Group: Baseline Data (T1) and Endline Data (T2).

		Group			*p*-Value
		Placebo	Curcumin	Curcumin + Piperine	ANCOVA/Bonferroni *
MPO	T1	11.5 ± 2.5	10.8 ± 2.6	10.7 ± 2.4	
	T2	11.8 ± 2.6	11.8 ± 1.6	12.9 ± 1.5	0.162
	Δ	−0.0 ± 3.7	1.0 ± 2.2	1.8 ± 1.9	0.199
TNF-α (pg/µL) ^Φ^	T1	0.0 ± 0.6	0.1 ± 0.6	0.2 ± 0.5	
	T2	0.0 ± 0.6	−0.4 ± 0.5	−0.2 ± 0.5	0.126
IL-17A (pg/µL) ^Φ^	T1	0.3 ± 0.2	0.1 ± 0.3	0.2 ± 0.4	
	T2	0.3 (0.2)	0.1 (0.5)	0.3 (0.2)	0.334 ^K^
IL 22 (pg/µL)	T1	3.4 ± 1.8	3.7 ± 2.0	3.9 ± 2.8	
	T2	4.2 ± 1.9	4.7 ± 1.7	5.1 ± 2.9	0.498
	Δ	1.3 ± 1.8	1.0 ± 2.1	1.0 ± 3.0	0.853
IL-10 (pg/µL)	T1	0.9 ± 0.7	1.0 ± 0.8	1.1 ± 0.8	
	T2	1.5 ± 0.7	1.9 ± 1.0	1.9 ± 1.2	0.511
	Δ	0.6 ± 0.7	0.8 ± 1.3	0.7 ± 1.3	0.940
SOD (U/µL)	T1	3599.8 ± 684.4	3728.1 ± 714.5	3761.3 ± 638.8	
	T2	3614.5 ± 731.5	3910.2 ± 800.4	4346.9 ± 879.0	0.020 ^##^
	Δ	−126.8 ± 762.7	142.1 ± 906.9	538.8 ± 1040.1	0.027 ^##^
Catalase (U/min)	T1	7.6 ± 2.9	8.5 ± 2.8	9.0 ± 3.2	
	T2	9.6 ± 4.2	9.2 ± 3.4	9.2 ± 3.4	0.781
	Δ	1.4 ± 6.1	0.6 ± 3.6	0.3 ± 2.6	0.576
Hydrogen Peroxide (nmol/mL)	T1	218.3 ± 155.7	224.7 ± 154.9	259.5 ± 149.7	
	T2	186.9 ± 118.7	215.5 ± 94.9	183.7 ± 98.1	0.307
	Δ	−45.7 ± 220.0	−9.1 ± 165.9	−78.3 ± 168.3	0.476
MDA (ng/µL) ^Φ^	T1	1.0 ± 0.2	0.9 ± 0.2	0.9 ± 0.2	
	T2	0.8 ± 0.4	0.7 ± 0.4	0.7 ± 0.3	0.602

Legend: Data expressed as mean ± Standard deviation or median (interquartile range); ^Φ^ = logarized variable; ^##^ = placebo group vs. curcumin + piperine group; ^K^ = Kruskal-Wallis test; pg = picogram; µL = microliter; mL = milliliter; U = unit; Δ = (T2–T1); ANCOVA adjusted for sex, age, and type of inflammatory bowel disease; * the Bonferroni test has been realized when ANCOVA < 0.05.

## Data Availability

This is an unpublished work which is not undergoing the submission process in any other scientific journal. All data are privately accessible.

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
