# Peer review of "Effect of Curcumin Plus Piperine on Redox Imbalance, Fecal Calprotectin and Cytokine Levels in Inflammatory Bowel Disease Patients: A Randomized, Double-Blind, Placebo-Controlled Clinical Trial"

_pharmaceuticals, 2024, doi:10.3390/ph17070849_

Round 1

Reviewer 1 Report

Comments and Suggestions for Authors

Manuscript ID: pharmaceuticals-3006527.

This study by da Paz Martins et al describes the effect of curcumin plus piperine in patients with Inflammatory Bowel Disease, specifically in redox imbalance

The strenght of the study is the novelty in the combination of both substances (curcumine and piperine) and the study of redox imbalance.

Nevertheless, there are some considerations the authors need to review.

MAJOR COMMENTS:

Abstract:

Consider to add definition of abbreviation of SOD.

Introduction:

It would be useful to add information related to the association of curcumin and piperine or explain why it was decided to associate both.

Results:

The first paragraph that appears after Results is the literal description that appears in the MDPI template. Please delete or modify for this article.

Consider explain the methods section before the results.

The explanation regarding physical activity is very scarce and is not recorded in the table nor  in the methods section. Please explain what kind of physical activity was recorded and the results in both groups with numbers (i.e. percentage).

Table 1 has three box with green color, please specify what does they mean.

The results of calprotectin levels exposed in table 4 could be more visual with a bar chart.

Aunque los efectos de la suplementación aparecen indicados en el material suplementario, en la parte que se explica aquí interesaría conocer el número de pacientes con pirosis en cada grupo (añadir esos datos numéricos).

As heartburn symptons are more common in the treatment group it is necessary to add numerical information (i.e. percentage) of patients with this digestive symptom (adverse event).

Methods:

- There are some patients that have IBD without any treatment at the momento of the inclusion. Please specify.

- It would be useful to give some explanation regarding the dose of curcumine and piperine chosen for this clinical trial.

- General date could be better explained in the section method: i.e., what means socioeconomics data and what means <4 school years (secondary school?). It would be useful to add questionarie as suplemmentary material.

- In pharmacological treatment another option no mesalazine or biologics is not recorded (i.e. azatioprine treatment). Please specify.

- Explain numbers regarding body max index (and different cut points in male and female).

- It would be interesting give visual information regarding ingredient characterization (cromatogram).

Discussion:

Give some references regarding this sentence: Studies indicate significant changes in SOD activity among IBD patients.

Systematic review and meta analysis of the group is cited twice. Please, unify.

Give more infomation regarding heartburn with curcumin. Add reference.

There are other limitations of the study:

- there is no a group with curcumin alone to check the beneficial effect of add piperine.

- mucosa inflamammation markers are no studied.

MINOR COMMENTS:

- Page 8: it seems a double space between ¨curcuminoids¨ and ¨chromatographics¨. Please, check it.

- Page 7: delete a dot between ¨marker¨  and ¨redox activity¨.

- Last paragraph of discussion is in italic letter, please modify

Author Response

Thank you for your careful reading and suggestions for modifications and additions to the manuscript.
All corrections were made, references included and text modified to better explain the results.
The answers one by one can be found in the attached file.
Once more, thank you
Regards

Reviewer 2 Report

Comments and Suggestions for Authors

the manuscript "Effect of Curcumin plus Piperine on Redox Imbalance and Inflammation in Inflammatory Bowel Disease Patients: a randomized, double-blind, placebo-controlled clinical trial" aimed to investigate the effect of curcumin plus piperine on specific parameters in IBD.

The title is scientifically nor biochemically correct. "Inflammation" should be appropriately explained. 

The abstract should be rewritten in a standard scientific way. The aim of the study should be clearly visible.

The text is full of different errors, including part of the template that remained included in the manuscript which, given the substantial number of authors (12!) raises the question if they all contributed to the manuscript or at least have read it:

"2. Results

This section was divided by subheadings. It should provide a concise and precise description of the experimental results, their interpretation, as well as the experimental conclusions that can be drawn" 

The number of patients if too low. Kindly calculate statistically the power of the study and kindly include more patients in the research.

Figure 1 should be adjusted technically for spacing and font

 Superoxide dismutase (SOD) activity - kindly mention the name of the kit manufacturer

The methodology is not appropriately described, and it is missing important information. The statistical analyses should be reconsidered, in first order, the power of the study should be absolutely calculated to find out the number of patients which have to be enrolled into the study in order to obtain reliable results.

Kindly reconsider the pretentious sentence in the summary "In summary, the findings of this study are unprecedent and highly relevant to the scientific community and individuals with IBD"

English language and grammar need revision. Technical and typographical errors as well.

Overall, the scientific contribution to the field is not significant.

Comments on the Quality of English Language

English style and grammar needs revision.

Author Response

Thank you for your consideration and care in reading the manuscript. All suggestions have been included, the English adjusted, and explanations added. We are grateful for the suggestions, however we believe that despite the reduction in final statistical power, caused by the loss of participants, the statistical strength of the final result on SOD justifies the publication and, mainly, the importance of the investigation for patients who have IBD and professionals who treat these patients.
All answers, point by point, are included in the annex

Best regards

Reviewer 3 Report

Comments and Suggestions for Authors

This study found that a 12-week supplementation of Curcumin plus piperine significantly increased serum superoxide dismutase (SOD) levels in patients with inflammatory bowel diseases (IBD) compared to a placebo group, suggesting potential therapeutic benefits in managing oxidative stress associated with IBD.  I think the paper can be published at the current format.

Below are some minor points.

1. Although the result has statistical significance, it would be beneficial for these authors to expand the cohort in future studies.

2. The results presented would be clearer if the authors could include some figures for illustration. (in addition to the concept figure)

Author Response

Thank you for your constructive feedback and the suggestion to expand the cohort in future studies. We appreciate your recognition of the statistical significance of our current findings. Indeed, we are planning to undertake a larger study to confirm and extend our results. However, we are currently constrained by limited research funding in our region of Brazil, which affects our ability to expand the cohort at this time. We are actively seeking funding opportunities to support this effort and are optimistic about conducting a more extensive study in the near future.

Several figures showing the action of SOD on Redox imbalance are already available, including the recent meta-analysis published by our group in this journal, which the authors understand would be very similar to the material already published.
Best regards